# $SL(2,\mathbb{C})$ Scheme Processing of Singularities in Quantum Computing and Genetics

**Michel Planat** [1,*,†] , **Marcelo M. Amaral** [2,†] , **David Chester** [2,†] **and Klee Irwin** [2,†]

1    CNRS, Institut FEMTO-ST, Université de Franche-Comté, F-25044 Besançon, France
2    Quantum Gravity Research, Los Angeles, CA 90290, USA
*    Correspondence: michel.planat@femto-st.fr
†    These authors contributed equally to this work.

**Abstract:** Revealing the time structure of physical or biological objects is usually performed thanks to the tools of signal processing such as the fast Fourier transform, Ramanujan sum signal processing, and many other techniques. For space-time topological objects in physics and biology, we propose a type of algebraic processing based on schemes in which the discrimination of singularities within objects is based on the space-time-spin group $SL(2,\mathbb{C})$. Such topological objects possess an homotopy structure encoded in their fundamental group, and the related $SL(2,\mathbb{C})$ multivariate polynomial character variety contains a plethora of singularities somehow analogous to the frequency spectrum in time structures. Our approach is applied to a model of quantum computing based on an Akbulut cork in exotic $R_4$, to an hyperbolic model of topological quantum computing based on magic states and to microRNAs in genetics. Such diverse topics reveal the manifold of possibilities of using the concept of a scheme spectrum.

**Keywords:** finitely generated group; $SL(2,\mathbb{C})$ character variety; algebraic surfaces; schemes; exotic $R_4$; topological quantum computing; microRNAs

**MSC:** 14E15; 14J70; 51H30; 81P68; 92D20; 57M05





## 1. Introduction

In signal processing of a time series, the lines of the Fourier spectrum are described by discontinuities. The approach may be generalized with number theory by taking Ramanujan sums as the building blocks of the signal expansion, e.g., see [1].

A more ambitious approach is to use algebraic geometry to reveal the singularities of an object. In recent papers, we explored topics of quantum computing and DNA biology with a common algebraic geometrical tool that we now call $SL(2,\mathbb{C})$ scheme processing. The group $SL(2,\mathbb{C})$ is found in physics in Einstein's field equations [2] and in loop quantum gravity [3]. For us, the necessary ingredient is a finitely generated group $\pi$ expressing the symmetries of the investigated object. The $SL(2,\mathbb{C})$ character variety [4] associated with $\pi$ is determined and summarized by its Groebner basis $\mathcal{G}$ [5,6]. The multivariate polynomials in $\mathcal{G}$ may contain isolated (or non-isolated) singularities that characterize the richness of the object. In this essay, the polynomials are reduced to surfaces living in the 3-dimensional projective space $P^3(\mathbb{Q})$ over the rationals.

Isolated singularities of a surface are easy to characterize and classify according to their A-D-E type in the McKay correspondence [7]. For a non zero dimensional singular subset in the surface, we may use tools of schemes for the resolution of singularities [8]. Scheme theory was created by A. Grothendieck to generalize smooth manifolds to algebraic varieties possibly decorated with singularities [9,10]. For our purpose, it is enough to see a scheme as a geometrical object defined by the vanishing of polynomials defined over an affine (or a projective space) like those living in $\mathcal{G}$.

Previously, our approach was restricted to algebraic surfaces containing isolated singularities in applications to topological quantum computing [11] and DNA-RNA transcription and regulation [12]. From now, the $SL(2, \mathbb{C})$ scheme processing is a good option to manage general singular sets. This is needed in our model of quantum computing based on an Akbulut cork in exotic $R_4$ [13], in our hyperbolic model of topological quantum computing based on magic states [14] and for some transcription factors and microRNAs in genetics [12].

In Section 2, we briefly describe the mathematical formalism used in our paper. It includes the definition of the character variety $\mathcal{V}$ representing the finitely generated group $\pi$ over the group $SL(2, \mathbb{C})$ and how a Groebner basis $\mathcal{G}$ is obtained from $\mathcal{V}$ in practice. The section mentions the distinction between simple singularities and singularities whose support is not zero dimensional. We then investigate the algebraic geometry of three types of complex objects in physics and biology. The first two objects rely on quantum computing following our previous papers [11,13,14]. Section 3 discusses the symmetries and related representations of the Akbulut cork $W$, a fundamental object in the theory of exotic 4-manifolds. Section 4 refers to the symmetries and the related representations of an hyperbolic 3-manifold found in the context of magic states in quantum computing. Section 5 investigates a third class of objects, which are a family of biological molecules called microRNAs that regulate the types and amounts of proteins [12]. In Section 6, we provide commentary on our results.

## 2. Theory

Details about the theory are described below. The corresponding implementation is performed on Sage [15] and Magma [16].

### 2.1. The $SL(2, \mathbb{C})$ Character Variety of a Finitely Generated Group and a Groebner Basis

Let $f_p$ be a finitely generated group; we describe the representations of $f_p$ in the (double cover alias the group extension of order two of the Lorentz group) $SL(2, \mathbb{C})$, the group of $(2 \times 2)$ matrices with complex entries and determinant 1. The group $SL(2, \mathbb{C})$ may be seen simultaneously as a 'space-time' (a Lorentz group) and a 'quantum' (a spin) group.

Representations of $f_p$ in $SL(2, \mathbb{C})$ are homomorphisms $\rho : f_p \to SL(2, \mathbb{C})$ with character $\kappa_\rho(g) = \text{tr}(\rho(g))$, $g \in f_p$. The set of characters allows to define an algebraic set by taking the quotient of the set of representations $\rho$ by the group $SL(2, \mathbb{C})$, which acts by conjugation on representations [17].

Such an algebraic set is called the $SL(2, \mathbb{C})$ character variety of $f_p$. It is made of a sequence of multivariate polynomials called a scheme $X$. The vanishing of polynomials defines the ideal $\mathcal{I}(X)$ of the scheme $X$. A Groebner basis $\mathcal{G}(X)$ is a particular set of the polynomial ring $\mathcal{I}(X)$ that has to follow algorithmic rules (similar to the Euclidean division for univariate polynomials).

For the effective calculations of the character variety, we make use of a software on Sage [15]. We also need Magma [16] for the calculation of a Groebner basis, at least for 3- and 4-letter sequences.

### 2.2. Singularities of an Algebraic Surface
Simple Singularities

The surfaces $S$ of interest in this case are said to be almost not singular in the sense that they have at worst simple singularities. In Magma, it is referred to a simple or A-D-E singularity [7] if it is an isolated singularity on $S$ that is analytically of the type $A_n$, $n \geq 1$, $D_n$, $n \geq 4$, $E_6$, $E_7$, or $E_8$.

The A-D-E type and the number of simple singularities are reflected in our notation. E.g., $S^{(lA_1)}$ means a surface with $l$ singularities of type $A_1$, $S^{(A_2)}$ means a surface with a single singularity of type $A_2$, and $S^{(D_4)}$ means a surface with a single singularity of type $D_4$. The Cayley cubic encountered in our previous paper is $\kappa_4^{(4A_1)}(x, y, z) = xyz + x^2 + y^2 + z^2 -$

4 [11]. The Fricke surface of type $D_4$ is $S^{(D_4)} = xyz + x^2 + y^2 + z^2 - 8(x + y + z) + 28$ ([18], Figure 16). Several other examples can be found in [12].

The relevant Magma command for such simply singular surfaces is *HasOnlySimpleSingularities(S)*.

### 2.3. Arbitrary Singularities

Let $X \in P^3(\mathbb{Q})$ be a singular surface (a surface containing a non zero dimensional singular subset of points). Let $Y \in P^3(\mathbb{Q})$ be a regular surface (devoid of singularities) above $X$, that is, a representation (in a sense to be qualified) $\rho : Y \to X$. The set of such morphisms $\rho$ belongs to the so-called spectrum $\text{Spec}_X(Y)$ of $Y$ in $X$.

In our context, the formal desingularization of an (hyper)surface $X$ in $P^3(\mathbb{Q})$ is realized with a proper birational map $Y \to X$, where $Y$ is regular. The formal desingularization is realized with the introduction of a formal prime divisor $\text{Spec } \widehat{\mathcal{O}_{X,p}} \to X$, where $p \in X$ is a regular point of codomension 1, $\mathcal{O}_{X,p}$ is the structure sheaf at $p$ of $X$ seen as a scheme, and the hat means the completion [19].

Taking the affine surface $S(x, y, z)$, the related homogeneous polynomial defines a hypersurface $X(x, y, z, w) \in P^3(\mathbb{Q})$ so that a formal prime divisor is actually a morphism $\phi : \text{Spec}_\phi[[t]] \to X$ defined by the $\mathbb{Q}$-algebra homomorphism $\phi^\sharp : \mathbb{Q}([x, y, z, w] / < X >\to \mathbb{F}_\phi[[t]]$, where $\mathbb{F}_\phi = \mathbb{Q}(s)[\alpha]$, $s$ is a parameter, and $\alpha$ is defined by the vanishing of a minimal polynomial.

The relevant Magma command for this case is *FormallyResolveProjectiveHyperSurface(S: AdjComp := true)*. Thanks to the setting *AdjComp := true*, the command returns the number of essential singularities allowing one to obtain the formal divisors needed to only compute birational invariants or adjoint spaces [19,20].

### 2.4. Kodaira–Enriques Classification

Given an ordinary projective surface $S$ in the projective space $P^3$ over a number field, if $S$ is birationally equivalent to a rational surface, the software Magma [16] determines the map to such a rational surface and returns its type within five categories. The returned type of $S$ is $P^2$ for the projective plane, a quadric surface (for a degree 2 surface in $P^3$), a rational ruled surface, a conic bundle, or a degree $p$ Del Pezzo surface where $1 \le p \le 9$ .

A further classification may be obtained for $S$ in $P^3$ if $S$ has at most point singularities (unless the singularities may be formally resolved as we described in the previous subsection for the case of characteristic zero). Magma computes the type of $S$ (or rather, the type of the non-singular projective surfaces in its birational equivalence class) according to the classification of Kodaira and Enriques [21]. The first returned value is the Kodaira dimension of $S$, which is $-\infty$, 0, 1, or 2. The second returned value further specifies the type within the Kodaira dimension $-\infty$ or 0 cases (and is irrelevant in the other two cases).

The Kodaira dimension $-\infty$ corresponds to birationally ruled surfaces. The second return in this case is the irregularity $q \ge 0$ of $S$. So $S$ is birationally equivalent to a ruled surface over a smooth curve of genus $q$ and is a rational surface if and only if $q$ is zero.

The Kodaira dimension 0 corresponds to surfaces that are birationally equivalent to a $K_3$ surface, an Enriques surface, a torus, or a bi-elliptic surface.

Every surface of the Kodaira dimension 1 is an elliptic surface (or a quasi-elliptic surface in characteristics 2 or 3), but the converse is not true: an elliptic surface can have Kodaira dimension $-\infty$, 0, or 1.

Surfaces of Kodaira dimension 2 are algebraic surfaces of a general type.

### A Singular Surface

Let us illustrate our approach with a selected singular surface encountered in the context of the transcription factor Prdm1 in our recent paper ([12], Section 3.1). The affine surface under question is $S_2(x, y, z) = z^4 + 2yz^3 + x^2 - 6yz - 2x - 8$. It contains 9 essential singularities in the desingularization. A 3-dimensional plot of $S_2(x, y, z)$ is in Figure 1. The surface is rational of the conic bundle type.

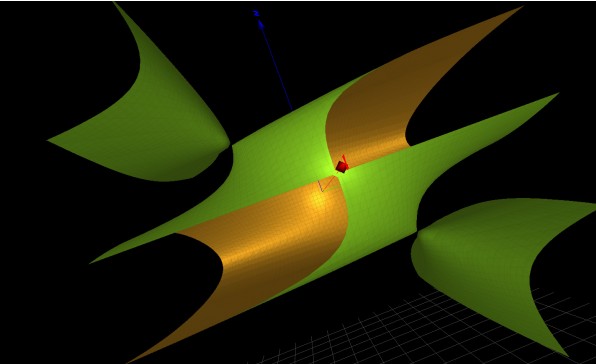

**Figure 1.** The affine singular surface $S_2(x, y, z) = z^4 + 2yz^3 + x^2 - 6yz - 2x - 8$ found in the Groebner basis for the transcription factor Prdm1 ([12], Section 3.1).

The projective surface is $S_2(x, y, z, w) = z^4 + 2yz^3 + w^2(x^2 - 6yz) - 2xw^3 - 8w^4$. There is 5 distinct types of minimal polynomial. For one of the divisors, the minimal polynomial is $\alpha^2 + 2s + 1$ and the prime divisor in the desingularization is the morphism $\phi : \text{Spec}_\phi[[t]] \to X$ defined by

$$x \to 1, \quad y \to st, \quad z \to t, \quad w \to \alpha t^2 + 0(t^4).$$

The 9 formal morphisms are used to compute the global sections (or adjoints) $\text{Hom}_{i,j}$. For $i, j < 3$, one obtains $\text{Hom}_{1,1} = \text{Hom}_{2,1} = \text{Hom}_{2,2} = \text{Hom}_{2,3} = \text{Hom}_{3,1} = \text{Hom}_{3,2} = \text{Hom}_{3,3} = 0$, $\text{Hom}_{1,2} = [z, w]$, $\text{Hom}_{1,3} = [xz, yz, z^2, xw, yw, zw, w^2]$.

### 3. $SL(2, \mathbb{C})$ Scheme Processing in Quantum Computing Based on an Akbulut Cork

*3.1. A Short Account of Magic States for Quantum Computing*

A quantum state is called magic when it is added to the eigenstates of the $d$-dimensional Pauli group to obtain universal quantum computation [22,23]. A subset of magic states consists of states associated with minimal informationally complete measurements, also called MIC states, see [14] and the references therein. In the context of a 3- or 4-manifold $\mathcal{M}$, of fundamental group $\pi_1(\mathcal{M})$, a subgroup of index $d$ of $\pi_1(\mathcal{M})$ has cosets organized according to a permutation group $\mathcal{P}_d$. Then, a permutation is also a permutation matrix and a possible source of MIC states arising from its eigenstates. Finally, for appropriate subgroups of $\pi_1(\mathcal{M})$, a finite geometry $\mathcal{F}_d$ is found from the two-point stabilizer subgroup of $\mathcal{P}_d$. The coset coordinates of $\mathcal{F}_d$ can be made in one to one correspondence with $d$-dimensional quantum observables. For example, important quantum geometries such as the Mermin $3 \times 3$ grid (at index $d = 9$), the generalized quadrangle of order two $GQ(2 \times 2)$ (at index $d = 15$), are encoding two-qubit contextuality [14] .

The Akbulut cork below belongs to the theory of 4-manifolds and features magic states of quantum computing [13]. To our earlier work, we add the investigation of $SL(2, \mathbb{C})$ character variety of the relevant fundamental groups. In particular, we put the Akbulut cork in a new perspective by revealing the singularity spectrum of its $SL(2, \mathbb{C})$ character variety.

*3.2. Brief Introduction to 4-Manifolds*

The theory of 4-manifolds is described in books [24–26]. Here, we are interested in the decomposition of a 4-manifold into one- and two-dimensional handles as shown in Figure 2 ([24], Figures 1.1 and 1.2). Let $B^n$ and $S^n$ be the $n$-dimensional ball and the $n$-dimensional sphere, respectively. An observer is placed at the boundary $\partial B^4 = S^3$ of the 0-handle $B^4$ and watches the attaching regions of the 1- and 2-handles. The attaching region of the 1-handle is a pair of balls $B^3$ (the yellow balls), and the attaching region of 2-handles is a framed knot (the red knotted circle) or a knot going over the 1-handle (shown in blue). For closed 4-manifolds, there is no need to visualize a 3-handle since it can be directly attached to the 0-handle. The 1-handle can also be figured out as a dotted circle

$S^1 \times B^3$ obtained by squeezing together the two three-dimensional balls $B^3$ so that they become flat and close together ([25], p. 169), as shown in Figure 2b.

For the attaching region of a 2- and a 3-handle one needs to enrich our knowledge by introducing the concept of an Akbulut cork to be described in the next paragraph ([13], Figure 3).

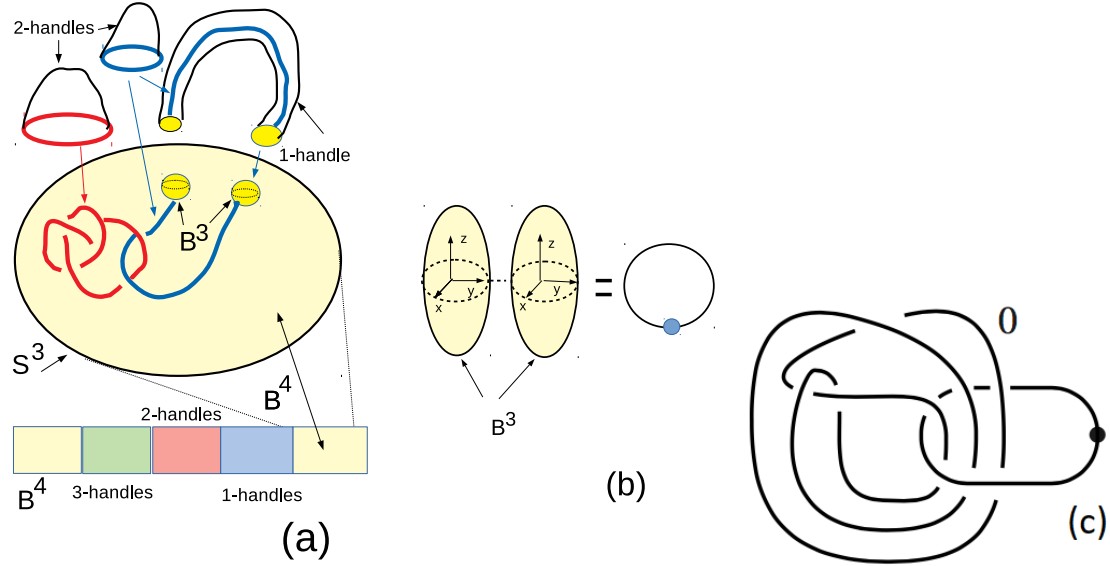

**Figure 2.** (**a**) Handlebody of a 4-manifold with the structure of 1- and 2-handles over the 0-handle $B^4$, (**b**) the structure of a 1-handle as a dotted circle $S^1 \times B^3$, and (**c**) an Akbulut cork $W = 9_{46}(-1, 1)$.

### 3.3. Akbulut Cork

A Mazur manifold is a contractible, compact, smooth 4-manifold (with boundary) not diffeomorphic to the standard 4-ball $B^4$ [24]. Its boundary is a homology 3-sphere. If we are restricted to Mazur manifolds that have a handle decomposition into a single 0-handle, a single 1-handle, and a single 2-handle, then the manifold has to be of the form of the dotted circle $S^1 \times B^3$ (as in Figure 2b) (right) union a 2-handle. The simplest object of this type is the Akbulut cork shown in Figure 2c [27,28].

Given $p, q, r$ (with $p \le q \le r$), the Brieskorn 3-manifold $\Sigma(p, q, r)$ is the intersection in the complex 3-space $\mathbb{C}^3$ of the 5-dimensional sphere $S^5$ with the surface of equation $z_1^p + z_2^q + z_3^r = 0$. The smallest known Mazur manifold is the Akbulut cork $W$, and its boundary is the Brieskorn homology sphere $\Sigma(2, 5, 7)$.

The Akbulut cork has a simple definition in terms of the framings $\pm 1$ of $(-3, 3, -3)$ pretzel knot also called $K = 9_{46}$ ([29], Figure 3). It has been shown that $\partial W = \Sigma(2, 5, 7) = K(1, 1)$ and $W = K(-1, 1)$.

An exotic $R^4$ is a differentiable manifold that is homeomorphic but not diffeomorphic to the Euclidean space $\mathbb{R}^4$. An exotic $R^4$ is called small if it can be smoothly embedded as an open subset of the standard $\mathbb{R}^4$ and is called large otherwise. Here, we are concerned with an example of a small exotic $R^4$.

According to [29], there is an involution $f : \partial W \to \partial W$ that surgers the dotted 1-handle $S^1 \times B^3$ to the 0-framed 2-handle $S^2 \times B^2$ and back, in the interior of $W$. There is a smooth contractible 4-manifold $V$ with $\partial V = \partial W$, such that $V$ is homeomorphic but not diffeomorphic to $W$ relative to the boundary ([27], Theorem 1). This leads us to our next paragraph.

### 3.4. The Manifold $\bar{W}$ Mediating the Akbulut Cobordism between Exotic Manifolds V and W

A cobordism between two oriented $m$-manifolds $M$ and $N$ is any oriented $(m + 1)$-manifold $W_0$ such that the boundary is $\partial W_0 = \bar{M} \cup N$, where $M$ appears with the reverse

orientation. The cobordism $M \times [0,1]$ is called the trivial cobordism. Next, a cobordism $W_0$ between $M$ and $N$ is called an h-cobordism if $W_0$ is homotopically like the trivial cobordism. The h-cobordism, according to S. Smale in 1960, states that if $M^m$ and $N^m$ are compact simply connected oriented $M$-manifolds that are h-cobordant through the simply-connected $(m+1)$-manifold $W_0^{m+1}$, then $M$ and $N$ are diffeomorphic ([26], p. 29).

However, this theorem fails in dimension 4. If $M$ and $N$ are cobordant 4-manifolds, then $N$ can be obtained from $M$ by cutting out a compact contractible submanifold $W$ and gluing it back in by using an involution of $\partial W$.

The h-cobordism under question in our example may be described by attaching an algebraic cancelling pair of 2- and 3-handles to the interior of Akbulut cork $W$. The 4-manifold $\bar{W}$ mediating $V$ and $W$ resembles the Akbulut cork with the dot replaced by a 0-surgery The manifold under question is nothing but $L7a6(0,1)(0,1)]$ (see [27], p. 355 or [13], Figure 3c).

*3.5. The Character Variety for an Akbulut Cork W*

The Sage code we used is as follows:
*from snappy import manifold*
*M=Manifold ('$9_{46}(-1,1)$')*
*G=M.fundamental_group()*
*I=G.character_variety_vars_and_polys (as_Ideal=True)*
*I*

The fundamental group ruling the Akbulut cork $W = 9_{46}(-1,1)$ is the two-generator group

$$\pi_1(W) = \left\langle a, b | aBAb^2 ABabaBABab, a^3 BAb^3 AB \right\rangle, \quad A = a^{-1}, \ B = b^{-1}. \tag{1}$$

With Sage software in Reference [15], or the aforementioned code, we compute the corresponding character variety. Then, from Magma [16], the Groebner basis is found in the form

$$\begin{aligned}
\mathcal{G}_W(x,y,z) &= z(z-2)[x+y+f_1(z)][y^2-y+f_2(z)][y+f_3(z)]f_4(z), \\
f_1(z) &= -171/34\, z^8 + 913/34\, z^7 - 2429/34\, z^6 + 4733/34\, z^5 - 2443/17\, z^4 \\
&\quad -224/17\, z^3 + 1847/17\, z^2 - 1103/34\, z - 1, \\
f_2(z) &= -95/34\, z^8 + 477/34\, z^7 - 1221/34\, z^6 + 2331/34\, z^5 - 1089/17 * z^4 \\
&\quad -285/17\, z^3 + 858/17\, z^2 - 337/34\, z - 1 \\
f_3(z) &= -71/17\, z^7 + 376/17\, z^6 - 997/17\, z^5 + 1942/17\, z^4 - 1993/17\, z^3 \\
&\quad -188/17\, z^2 + 1477/17\, z - 436/17, \\
f_4(z) &= z^7 - 5\, z^6 + 13\, z^5 - 25\, z^4 + 24\, z^3 + 4\, z^2 - 18\, z + 5.
\end{aligned} \tag{2}$$

The factor containing $f_1(z)$ in Equation (3) is the singular surface $S_W(x,y,z)$ shown in Figure 3. It is a rational scroll and a surface of a general type. The factor containing $f_2(z)$ is an hyperelliptic function of genus 7, discriminant $\approx 1.327502101$ and points at infinity $(1,0,0)$ and $(1,-95/34,0)$. The factor containing $f_3(z)$ is an ordinary curve. The factor containing $f_4(z)$ is a seventh-order polynomial.

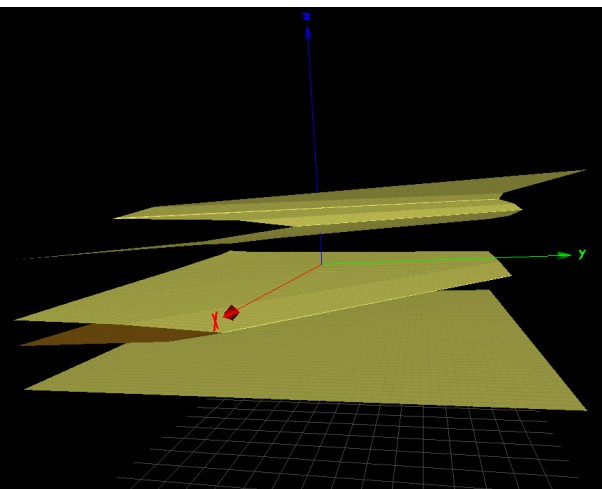

**Figure 3.** The surface $S_W(x, y, z)$ found within the Groebner basis of the $SL(2, \mathbb{C})$ character variety for the Akbulut cork $W$.

Formal Desingularization of the Surface $S_W(x, y, z)$

The formal desingularization of a hypersurface $X$ in the three-dimensional projective space $\mathbb{P}^3_{\mathbb{Q}}$ over the rationals $\mathbb{Q}$ is described in [19] and can be explicitely given with Magma ([16], Section 122.5.3).

To the surface $X = S_W(x, y, z)$, we associate the degree 8 homogeneous polynomial $S_W(x, y, z, w) = x + y + f_1(z, w) \in \mathbb{Q}(x, y, z, w)$, with $f_1(z, w) = -171/34 z^8 + 913/34\, wz^7 + \cdots - 1103/343\, w^7 z - w^8$.

For the projective surface $X = S_W(x, y, z, w)$, the two essential (over the three) singular morphisms are

$$
\begin{aligned}
x &\to 1 \\
y &\to st - 1 \\
z &\to t \\
w &\to \alpha t + O(t^8),
\end{aligned}
\tag{3}
$$

$$
\begin{aligned}
x &\to 1 \\
y &\to s \\
z &\to 34/171\,(s+1)t^7 + O(t^8) \\
w &\to O(t^8),
\end{aligned}
\tag{4}
$$

where $\alpha$ has minimal polynomial $-s - \alpha^8 f_1(\alpha^{-1})$.

These formal morphisms are used to compute the global sections (or adjoints) $\mathrm{Hom}_{i,j}$. For the surface $S_W(x, y, z)$, one gets
$\mathrm{Hom}_{1,1} = \mathrm{Hom}_{2,1} = \mathrm{Hom}_{2,2} = 0$, $\mathrm{Hom}_{1,2} = [z^6, z^5 w, z^4 w^2, z^3 w^3, z^2 w^4, z w^5, w^6] \cdots$.

*3.6. The Character Variety for the Mediating Manifold $\bar{W}$*

The fundamental group $\pi_1(\bar{W})$ of the h-cobordism $\bar{W}$ is as follows: [13]

$$
\pi_1(\bar{W}) = \left\langle a, b \,|\, a^3 b^2 AB^3 Ab^2, (ab)^2 aB^2 Ab^2 AB^2 \right\rangle.
\tag{5}
$$

The cardinality structure of subgroups of this fundamental group is

$$
\eta_d[\pi_1(\bar{W})] = \eta_d[\pi_1(W)] = [1, 0, 0, 0, 1,\ 1, 2, 0, 0, \mathbf{1},\ 0, \mathbf{5}, 4, \mathbf{9}, 7,\ 1 \cdots],
$$

where the bold digits refers to our investigation in ([13], Table 1) concerning the existence of a finite geometry obtained with a subgroup of the corresponding index. The

smallest case occurs at index 10, and the geometry is the Mermin pentagram, a type of contextual geometry.

As before, with Sage software in Reference [15], we compute the corresponding character variety. Then, from Magma, the Groebner basis is found in the form

$$
\begin{aligned}
\mathcal{G}_{\bar{W}} &= (z-2)(z^2 - z - 1)[x + g_1(z)][y + g_2(z)]g_3(z), \\
g_1(z) &= -z^9 + 3z^8 + 5z^7 - 18z^6 - 10z^5 + 43z^4 - 2z^3 - 27z^2 + 3z + 4, \\
g_2(z) &= -z^8 + 2z^7 + 6z^6 - 11z^5 - 15z^4 + 24z^3 + 9z^2 - 10z - 2, \\
g_3(z) &= z^7 + z^6 - 5z^5 - 6z^4 + 6z^3 + 5z^2 - 2z - 1.
\end{aligned}
\tag{6}
$$

Although the card seq for manifolds $W$ and $\bar{W}$ are the same, we clearly see that the character varieties are distinct. In the later case, it contains two ordinary curves and polynomials of degrees 1, 2, and 7.

## 4. $SL(2, \mathbb{C})$ Scheme Processing in Topological Quantum Computing

In this section, we are interested in the $SL(2, \mathbb{C})$ character variety of the fundamental group of (hyperbolic) 3-manifold L10n46 (alias otet08$_{00002}$). This manifold describes the 4-fold (irregular) covering of the figure-eight knot $4_1 = K4a1$ ([14], Table 2). It is connected to the magic state describing the contextual geometry of two-qubits (e.g., [14] Figure 1 or [30] Table 1).

Thanks to the character variety and the related algebraic surfaces , we can add the topological aspect to the previous description.

The Groebner basis of the character variety for the fundamental group of 3-manifold L10n46 may be obtained with Magma. The 3-generator fundamental group is

$$
\pi_1(\text{L10n46}) = \langle a, b, c \,|\, abAcBac, abbCBccBBBACbcbC \rangle.
\tag{7}
$$

The character variety $\mathcal{G}_{\text{L10n46}}(e, f, g, h)(x, y, z)$ is now seven-dimensional as in [12], and the selected choices of parameters $e$, $f$, $g$, and $h$ allow us to determine the algebraic surfaces within the Groebner basis.

For instance, we find

$$
\begin{aligned}
\mathcal{G}_{\text{L10n46}}(0,0,0,0) &= y S_1(x,y,z) S_2(x,y,z) S_3(x,y,z) \cdots \\
S_1(x,y,z) &= f_{2,\{\}}^{(A_1)}(x,y,z) = xyz + xz^2 + x^2 + y^2 + z^2 + yz - x - 6, \\
S_2(x,y,z) &= xz^2 + yz - x - 2, \\
S_3(x,y,z) &= -x^2z^2 + 2x^2 + y^2 + z^2 + 2x - 4.
\end{aligned}
\tag{8}
$$

The first surface $S_1(x,y,z) = f_{2,\{\}}^{(A_1)}$ in the product (9) has a single simple singularity of type $A_1$ whose reduced singular subscheme is of degree 2 with a vanishing support.

The second surface $S_2(x,y,z)$ is a singular rational scroll with a single essential singularity. The corresponding singular morphism $\text{Spec}\, \mathbb{F}_\phi[[t]] \to X = S_2(x,y,z,w)$ is

$$
\begin{aligned}
x &\to 1 \\
y &\to s \\
z &\to t \\
w &\to ((-2s^3 - 6s)/(s^2 + 4)\alpha + (-2s^2 - 4)/(s^2 + 4))\alpha t + O(t^3),
\end{aligned}
\tag{9}
$$

where $\alpha$ has minimal polynomial $\alpha^2 - s\alpha - 1$.

The third surface $S_3(x,y,z) = -x^2z^2 + 2x^2 + 2x + y^2 + z^2 - 4$ is a singular surface of the degree 4 del Pezzo type. There are 4 essential singularities. We retain one of the two

singularities with a nontrivial minimal polynomial. The corresponding singular morphism $\text{Spec } \mathbb{F}_\phi[[t]] \to X = S_1(x, y, z, w)$ is

$$
\begin{aligned}
x &\to 1 \\
y &\to s \\
z &\to t \\
w &\to \alpha t - 1/(s^4 + 4s^2 + 4)t^2 + (-1/2\,s^4 + 9/2)/(s^6 + 6s^4 + 12s^2 + 8)\alpha t^3) + O(t^5)
\end{aligned}
\tag{10}
$$

where $\alpha$ has minimal polynomial $\alpha^2 - 1/(s^2 + 2)$.

All three affine surfaces are shown in Figure 4.

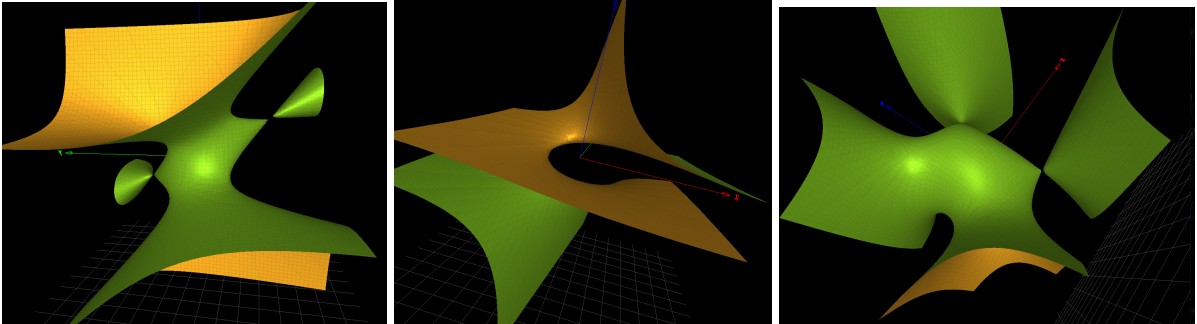

**Figure 4.** Left: the (degree 3) del Pezzo surface $S_1(x, y, z) = f_{2,\{\}}^{(A_1)}(x, y, z) = xyz + xz^2 + x^2 + y^2 + z^2 + yz - x - 6$. Middle: the (rational scroll) surface $S_2(x, y, z) = xz^2 + yz - x - 2$. Right: the (del Pezzo degree 4) surface $S_3(x, y, z) = -x^2z^2 + 2x^2 + y^2 + z^2 + 2x - 4$.

## 5. $SL(2, \mathbb{C})$ Scheme Processing in microRNAs

In this section, we focus on the $SL(2, \mathbb{C})$ character varieties attached to microRNAs (miRNAs for short). This case was already tackled in our recent paper [12].

The miRNAs are short (approximately 22 nt long) single-stranded RNA molecules playing a fundamental role in the expression and regulation of genes by targeting specific messenger RNAs (mRNAs) for degradation or translational repression. The genes encoding miRNAs are much longer than the processed mature miRNA molecule. There are pre-miRNAs, approximately 70-nucleotides in length, that are folded into imperfect stem-loop structures.

Each miRNA is synthesized as an miRNA duplex comprising two strands (-5p and -3p). However, only one of the two strands becomes active, which is selectively incorporated into the RNA-induced silencing complex in a process known as miRNA strand selection [31,32]. For details about the miRNA sequences, we use the Mir database [33,34].

Disregulation of miRNAs may lead to a disease such as cancer. A key microRNA known as an oncomir (involved in immunity and cancer) is mir-155 [12].

Here, we select examples of human miRNAs from the perspective of evolution. The generator of the group $\pi$ to be considered is a short (about 8-letter long) seed made of two to four distinct bases in the set {A,U,G,C}. Most of the time, $\pi$ is close to a free group of rank equal the number of distinct bases in the seed minus one. This point can be checked by the cardinality structure of conjugacy classes subgroups of $\pi$ (denoted card seq). Exceptions to this rule and the occurrence of singularities (isolated or not) in the corresponding character variety built from $\pi$ are a witness of a potential disease. Unlike the case of transcription factors, until now, singularities possibly found with microRNAs are isolated singularities.

According to Reference ([35], Table 3), the slowest evolving miRNA gene is hsa-mir-503 (the notation hsa is for the human specie). It is known that mir-503 regulates gene expression from different aspects of pathological processes of diseases, including carcinogenesis, angiogenesis, tissue fibrosis, and oxidative stress [36]. The seed region for mir-503-5p is AGCAGCGG, and the corresponding Groebner basis for parameters

$(e, f, g, h) = (0, 0, 0, 0)$ is very simple: $\mathcal{G}_{mir-503-5p}(0,0,0,0) = \kappa_4^{(4A_1)}(x, y, z)$, as shown in Figure 5 (Left).

For $(e, f, g, h) = (1, 1, 0, 0)$, $\mathcal{G}_{mir-503-5p}(1, 1, 0, 0) = -3xyz\kappa_3(x, y, z)$, and $\kappa_3(x, y, z)$ is the Fricke surface found in ([11], Section 3.3). For $(e, f, g, h) = (1, 1, 1, 1)$, there are many more polynomials. One of them defines the Fricke surface $xyz + x^2 + y^2 + z^2 - 2x - y - 2$. The considered seed region for mir-503-3p is GGGUAUU. The surfaces in the Groebner basis are very simple in this case, and not even simple singularities lie in them.

One of the fastest evolving microRNA is mir-214 ([35], Table 3). First mir-214 was reported to promote apoptosis in HeLa cells. Presently, mi-214 is implicated in an extensive range of conditions such as cardiovascular diseases, cancers, bone formation, and cell differentiation [37]. For mir-214-5p and the seed sequence GCCUGU, one finds the surface $f_{1,\{1:0:0:0\}}^{(A2)}(x, y, z) = xyz + y^2 + z^2 - 4$ within $\mathcal{G}_{mir-214-5p}(0, 0, 0, 0)$, as shown in Figure 5. A singular surface with the same support $f(x, y, z) = xyz + y^2 + z^2 - y - 2z - 1$ is found in $\mathcal{G}_{mir-214-5p}(1, 1, 1, 1)$. For a longer seed, surfaces are not found to contain isolated singularities.

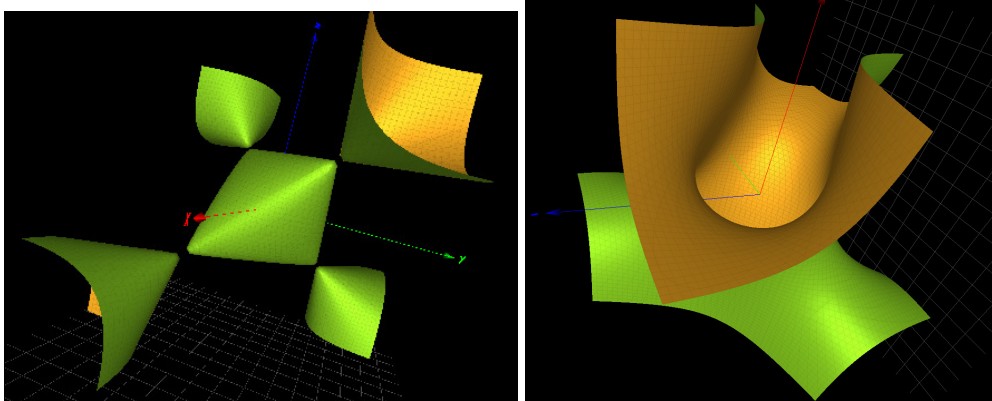

**Figure 5.** Left: the Cayley cubic $\kappa_4(x, y, z)$ found in the character variety for the slowest evolving miRNA gene hsa-mir-503. The surface $f_{1,\{1:0:0:0\}}^{(A2)}(x, y, z)$ found in the character variety $\mathcal{G}_{mir-214-5p}$ of the fast evolving gene hsa-mir-214.

## 6. Conclusions

Over the last few years, the authors of this article have found that some mathematical techniques employed for quantum information processing and quantum computing may also apply to biology at the genome scale. More precisely, group theory and representations of symmetries with characters of finite groups have been used for topological quantum computing (TQC) [14] or elementary particles [38], and the encoding of proteins [39]. Methods for dealing with infinite groups and $SL(2, \mathbb{C})$ representations of such groups in TQC papers [11,40] were similarly employed for transcription factors [12] and miRNAs [12].

Our efforts in this paper belong to the field of scheme processing, where the desingularization of discontinuities of algebraic surfaces is a spectrum. The scheme spectrum is a well known concept in commutative algebra. The prime spectrum of a ring is the set of prime ideals of the ring $R$ and is denoted by $\mathrm{Spec}(R)$. The sheaf of rings $\mathcal{O}$ is the relevant algebraic geometrical notion introduced by Grothendieck to develop this field [8,41]. We touched on this important concept of schemes while investigating non-zero dimensional sets of singularities in surfaces belonging to the $SL(2, \mathbb{C})$ character variety of an infinite group. We are fortunate that the software Magma is designed to implement schemes in a variety of applications (curves, surfaces, and more). Another computer algebra system with similar facilities is Singular [42,43].

We have no doubt that scheme theory will play and increasing role in other fields such as cosmology [44] and the field of 'quantum consciousness' [45]. One possible field of application is the so-called 'quantum consciousness' [45].

**Author Contributions:** Conceptualization, M.P.; methodology, M.P. and M.M.A.; software, M.P.; validation, M.M.A., D.C. and K.I.; formal analysis, D.C. and M.P.; investigation, D.C., M.M.A. and M.P.; resources, K.I.; data curation, M.P.; writing—original draft preparation, M.P.; writing—review and editing, M.P. and M.M.A.; visualization, M.M.A.; supervision, M.P.; project administration, M.P. and K.I.; and funding acquisition, K.I. All of the authors have read and agreed to the published version of the manuscript.

**Funding:** This research received no external funding.

**Informed Consent Statement:** Not applicable.

**Data Availability Statement:** Data are available from the authors after a reasonable demand.

**Conflicts of Interest:** The authors declare no conflict of interest.

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
