# Peer review of "SL(2,C) Scheme Processing of Singularities in Quantum Computing and Genetics"

_axioms, doi:10.3390/axioms12030233_

Round 1

Reviewer 1 Report

In paper Authors presents several examples of SL(2,C) scheme applied to three types of manifolds. Generally speaking, paper possesses a correct general structure. After introduction and description of necessary theory Authors present the main examples of manifolds and  try to point out additional applications of manifolds in sections 2, 3, 4.

From general point of view discussion about manifolds and its relation to topics in quantum computation model and genetics seems to very interested and important. But, in this paper presentation of application of manifolds to mentioned computation models and area of science is very limited. 

It should be pointed in general the paper is short and contains only 10 pages. But, the title suggests that Authors show (give/outline) the applications of presented scheme for quantum computing and genetics. But, the presentation of manifolds to quantum computing (QC) is very limited -- only few lines from 229 to 232 are referred to some items from bibliography. The rest of section 3 contains description of scheme processing. Therefore, in my opinion the title of contribution should be changed or Authors should present more detail about relation of discussed problem to quantum computational methods. The section 4 contains bigger discussion about application presented scheme to mRNA processing. Similar example should be also given for quantum computation.

I would like also to add that some part of paper should be also improved, e.g.

+ Authors mentioned that they used some functions from Magma/Sage to compute some aspects of discussed problem, but this code is not directly available,  access to code repository with all computational example will improve final score of paper and allows to verify discussed problems,

+ paper has some technical issues with figures, Authors should reconsider the change of the black background colour  with a white background, then change of the colour of plot to black, which should improve the readability,

+ the introduction section is enumerated from zero.

Therefore, in my opinion paper should be reviewed again, after necessary technical corrections and adding some remarks about application of presented scheme in area of quantum computation model or general topological quantum computations (this model is mentioned at abstract of reviewed paper). Adding source code repository with some additional comments in paper and examples should also improved the final score of paper.

Author Response

// Response to Referee 1.

We thank you for your report and most of your comments are justified, in our opinion. We tried to account for all of them.

1. The relevance to quantum computing is explained in Sections 2 and 3. We agree that the connection of the Akbulut cork to quantum computing was not 
made explicit enough. For this reason, we added a paragraph at the beginning of Section 2 " 3.1. A short account of magic states for quantum computing" 
and we reorganized the subsections. As a result, the title of our paper is fully justified. 

2. We gave proper reference to the code employed for calculating the SL(2,C) character variety. Ref. 13 is freely available. 
But to follow your suggestion, we added the simple lines of code for obtaining the character variety of the Akbulut cork at the beginning of Section 3.5.

3. The presentation in the introduction has been improved. We added several references about the importance of the SL(2,C) group in physics, the definition
 of a character variety and of a Groebner basis.

4. We tried to redraw the figures in black with a white background, but the result is not  satisfactory. So we skipped this item.

5. We corrected the enumeration.

6. Eleven  external references have been added and the number of self-citations has been reduced to 20%  to follow MDPI's citation policy.

The changes are highligthted in the pdf file.

Reviewer 2 Report

This paper continues a series of works of the authors devoted to introduce highly nontrivial mathematical techniques that can be relevant in some applications, such as quantum information processing, quantum computing, and biology. 

More specifically, in this paper they use the notion of schemes, due to Grothedieck. This is a sound but difficult notion. I should recognize that I am not an expert in the field and I had a hard work trying to understand the paper. From my previous experience with papers of these authors (mainly Michel Planat), I am confident the results are correct. However, I always have the same problem: these sophisticated concepts explain some phenomena in a very compelling manner. However, in which way they contribute to new knowledge? In other words, these notions allows one to understand things that they are already well understood by means of other (more pedestrian) methods. Apart from elegance, do they contribute to the advancement of the field.

I think that is a crucial question that the authors should answer in an appropriate manner. 

Author Response

Thank you for the report about our paper.

We performed several changes in the presentation to better expain our approach and the connection to quantum computing.
More precisely

1. The relevance to quantum computing is explained in Sections 2 and 3. The connection of the Akbulut cork to quantum computing was not 
made explicit enough. For this reason, we added a paragraph at the beginning of Section 2 " 3.1. A short account of magic states for quantum computing" 
and we reorganized the subsections. 

2. We added the simple lines of code for obtaining the character variety of the Akbulut cork at the beginning of Section 3.5.

3. The presentation in the introduction has been improved. We added several references about the importance of the SL(2,C) group in physics, the definition
 of a character variety and of a Groebner basis.

4. Eleven external references have been added and the number of self-citations has been reduced to 20%  to follow MDPI's citation policy.

The changes are highligthted in the pdf file.

Reviewer 3 Report

This is a great paper, I suggest to accept as is.

Author Response

//  Response to  Referee 3

Thank you the reading our paper.
We are glad of your appreciation.
To follow the comments of the other 3 three referees, we did several changes to improve the presentation.  
The changes are highligthted in the pdf file.

Reviewer 4 Report

It is not clear what is the aim of the paper and what is the novelty of it.

The authors have proposed SL(2,C) schemes, but it is not known what their novelty is, because the Authors refer to their previous articles. Sometimes the reader has the impression that the article is a review of the literature, with particular emphasis on earlier articles by these authors.

The authors refer to the commercial software they used. They used the strict procedures of this software. What was the contribution of their work in research using this software.

A more detailed review, taking into account the details of the paper editting, will be possible after the Authors provide the purpose of the work, the novelty of the results obtained and specific applications.

Author Response

Thank you for your comments. However we are surprised by your opinion. 
They may be motivated by an imperfect presentation of our results.
We have improved the presentation. in particular 

1. The relevance to quantum computing is explained in Sections 2 and 3. The connection of the Akbulut cork to quantum computing was not 
made explicit enough. For this reason, we added a paragraph at the beginning of Section 2 " 3.1. A short account of magic states for quantum computing" 
and we reorganized the subsections. 

2. We added the simple lines of code for obtaining the character variety of the Akbulut cork at the beginning of Section 3.5.

3. The presentation in the introduction has been improved. We added several references about the importance of the SL(2,C) group in physics, the definition
 of a character variety and of a Groebner basis. To be more precise about our progress and the novelty of our work we reorganized the introduction.
 What is new compared to our previous work is now emphasized. 

4. Eleven  external references have been added and the number of self-citations has been reduced to 20%  to follow MDPI's citation policy.

The changes are highligthted in the pdf file.

Round 2

Reviewer 1 Report

In paper Authors made several changes into the article text which answered for raised problems outlined in the first review. Also some other technical issues have been corrected, however technical quality of Figure 1, black background should be still corrected.

Other methodological changes  e.g. small example of code (line 216) and longer comments about quantum magic states (line 144), improve the overall scientific score of paper. Therefore, in my opinion paper could be published.

Reviewer 2 Report

The authors have considered in detail all the points in my report. Although the paper, by its very nature, is hard to read, it has considerably improved. I recommend publication.

Reviewer 4 Report

The Authors has added some information and reduced selfcitations.